# Characteristic Analysis of AlGaN/GaN HEMT with Composited Buffer Layer on High-Heat Dissipation Poly-AlN Substrates

**DOI:** 10.3390/membranes11110848

**Published:** 2021-10-30

**Authors:** Chong-Rong Huang, Hsien-Chin Chiu, Chia-Hao Liu, Hsiang-Chun Wang, Hsuan-Ling Kao, Chih-Tien Chen, Kuo-Jen Chang

**Affiliations:** 1Department of Electronics Engineering, Chang Gung University, Taoyuan 333, Taiwan; gain525252@gmail.com (C.-R.H.); r3287133@gmail.com (C.-H.L.); smallflgt@hotmail.com (H.-C.W.); snoopy@mail.cgu.edu.tw (H.-L.K.); 2Department of Radiation Oncology, Chang Gung Memorial Hospital, Chang Gung University, Taoyuan 333, Taiwan; 3National Chung-Shan Institute of Science and Technology, Materials and Electro-Optics Research Division, Taoyuan 333, Taiwan; greatdc38@gmail.com (C.-T.C.); mike.ckj@gmail.com (K.-J.C.)

**Keywords:** QST substrate, back-barrier layer, high thermal conductivity

## Abstract

In this study, an AlGaN/GaN high-electron-mobility transistor (HEMT) was grown through metal organic chemical vapor deposition on a Qromis Substrate Technology (QST). The GaN on the QST device exhibited a superior heat dissipation performance to the GaN on a Si device because of the higher thermal conductivity of the QST substrate. Thermal imaging analysis indicated that the temperature variation of the GaN on the QST device was 4.5 °C and that of the GaN on the Si device was 9.2 °C at a drain-to-source current (*I*_DS_) of 300 mA/mm following 50 s of operation. Compared with the GaN HEMT on the Si device, the GaN on the QST device exhibited a lower *I*_DS_ degradation at high temperatures (17.5% at 400 K). The QST substrate is suitable for employment in different temperature environments because of its high thermal stability.

## 1. Introduction

GaN is widely used in high-frequency and high-power next-generation devices because of its two-dimensional electron gas (2DEG) concentration, high carrier mobility, low ON resistance, and high breakdown voltage [1,2,3]. GaN has demonstrated increasing potential for a wide range of applications. Sapphire and Si are commonly used as substrate materials for GaN; however, their low thermal conductivity limits heat dissipation from device-level self-heating during the operation of high-electron-mobility transistors (HEMTs) and may influence the electrical characteristics, reliability, and performance of HEMTs [4,5,6]. Thus, for most applications, replacement substrates such as SiC or GaN are used to improve the device performance; however, their high cost is problematic. The poly-aluminum nitride (AlN) substrate (QST) is promising for GaN-based HEMTs because of its high thermal dissipation efficiency and high mechanical strength.

Another key concern is the large lattice mismatch between GaN and substrates. Currently, the lattice mismatch in buffer layers is compensated with Fe and C doping, which causes the semi-insulating layer to increase the breakdown voltage and reduce the leakage current of the device. However, the Fe-doped buffer layer may have memory effects of the Fe diffusion associated with high growth temperatures [7,8,9], whereas severe current collapse can result from the trapping effects related to deep acceptors in the C-doped buffer layer [10,11,12]. In this study, a back-barrier (BB) layer was added to the buffer layer to reduce the influence of the doped acceptor between the channel and buffer layers. This composite buffer layer increased the withstand voltage of the relevant fabricated device and reduced the current collapse effect. In addition, because of the different thermal conductivities of AlGaN and GaN [13], the AlGaN BB layer influences the substrate heat dissipation capability, which is also a key index for radiofrequency (RF) and power device applications [14]. Therefore, this study added the composite buffer layer on a high-heat-dissipation QST substrate to reduce the low-heat-dissipation effect caused by the AlGaN BB layer. Because of the high thermal conductivity of the QST core, the thermal resistance of a QST substrate is lower than that of a Si substrate, and the QST substrate can reduce the effect of heat on a device [15].

## 2. Device Structure

The epitaxial layers for Al_0.24_Ga_0.76_N/GaN RF HEMTs were grown through metal organic chemical vapor deposition (MOCVD) on high-thermal-conductivity QST and Si substrates. Prior to the preparation of the buffer and active layers, an AlN nucleation layer was grown to compensate for the lattice mismatch and reduce the dislocation density in the fabricated devices. Then, a high-isolation Fe- or C-doped GaN buffer layer was prepared. A 50-nm Al_0.05_Ga_0.95_N BB layer was first grown to increase the conduction band energy in the buffer layer and reduce the leakage current. Subsequently, a 300-nm GaN channel layer was prepared. A 0.5-nm-thick AlN spacer layer contributed to the reduction in interface roughness and enhanced the carrier mobility of a 2DEG, and an 18-nm-thick Al_0.24_Ga_0.86_N barrier layer was deposited. Finally, a 2-nm GaN cap layer was deposited through MOCVD to reduce the surface oxidation and leakage current of the AlGaN barrier layer. The HEMTs with high thermal dissipation efficiency and mechanical strength were fabricated on a poly-AlN substrate (Figure 1).

Device fabrication began with mesa isolation by using an inductively coupled plasma system with a BCl_3_ + Cl_2_ mixed gas in a reactive ion etching chamber. Subsequently, a metal film of ohmic contacts was prepared through electron beam evaporation (e-Gun) with multilayered Ti/Al/Ni/Au (25, 130, 25, and 80 nm, respectively; drain-to-source distance L_DS_ = 4 µm). To form an ohmic contact, we annealed the corresponding device at 875 °C for 30 s in a nitrogen-rich environment by using a rapid thermal annealing system. In the gate process, a Schottky gate (device gate length *L_G_* = 1 µm; gate width *W_G_* = 50 µm) was defined through electron beam evaporation, and the gate metal was formed using Ni/Au (25 and 80 nm, respectively). A metal film of interconnected Ti/Au (25 and 80 nm, respectively) was deposited to reduce the contact resistance. Finally, a 100-nm Si_3_N_4_ passivation layer was deposited through plasma-enhanced chemical vapor deposition.

Figure 2 presents the X-ray diffraction (XRD) results of GaN on QST and Si. The (002) and (102) XRD profiles cover angles of 15°–19° and 22°–27°, respectively, along the omega axis. Furthermore, the full width at half maximum (FWHM) values for the (002) and (102) planes of the QST and Si devices were 628/673 and 1244/897 arcsec, respectively. The total dislocation of GaN on QST was 9.01 × 10^9^/cm^2^ and that of GaN on Si was 5.18 × 10^9^/cm^2^. The dislocation was calculated using the XRD FWHM results as follows [16,17]:(1)$$N_{screw}=\frac{FWHM_{002}^{2}}{4.35\times b_{screw}^{2}},N_{edge}=\frac{FWHM_{102}^{2}}{4.35\times b_{edge}^{2}}$$
(2)$$N_{total}=N_{screw}+N_{edge}$$
where *N_screw_* and *N_edge_* are the screw and edge dislocation density, respectively, and *b* is the Burger’s vector.

## 3. Experimental Result and Discussion

To study the effect of the QST substrate on the device performance, we measured the drain-to-source current (*I*_DS_)–gate-to-source voltage (*V*_GS_), *I*_DS_–*V*_DS_, and *I*_GS_–*V*_GS_ characteristics of the fabricated devices using an Agilent 4142B semiconductor parameter analyzer (Figure 3). The drain-to-source current (*I*_DS_) and output transconductance (*g*_m_) versus gate-to-source voltage (*V*_GS_) at *V*_DS_ were 10 V for a *V*_GS_ sweep of −6 to 2 V. The saturation currents of GaN on Si and QST were 543 and 545 mA/mm, respectively, at a *V*_GS_ value of 2 V and a *V*_DS_ value of 10 V. Figure 3b depicts the comparison of the ON-resistance (*R*_ON_) values for the devices with GaN on Si and QST, which were 4.44 (*R*_ON_Si_) and 5.2 Ω·mm (*R*_ON_QST_), respectively. As illustrated in Figure 3c, the Si device had the best *I*_on_/*I*_off_ ratio and subthreshold swing (SS) of 5.57 × 10^4^ and 0.19 V/dec, respectively. The *I*_on_/*I*_off_ ratio and SS of the QST device were 2.38 × 10^3^ and 0.57 V/dec, respectively. The results revealed that the total dislocation of the GaN on the QST device was greater than that of the GaN on the Si device; thus, the GaN on the QST device had a higher drain-to-source leakage current (*I*_DS_) at a *V*_GS_ of −6 V. Figure 3d presents the small-signal measurements of the QST, and Si devices collected using an Agilent network analyzer. These measurements indicated that the maximum current gain (*f*_T_) and maximum power gain (*f*_max_) of the Si device were 4.5 and 11.6 GHz, respectively. The *f*_T_ and *f*_max_ values of the QST device were 7.2 and 9.1 GHz, respectively.

To explore the influence of ambient temperature on the device characteristics, we performed variable-temperature measurements on the two devices. Figure 4a illustrates the *I*_DS_–*V*_GS_ characteristics measured from 300 to 400 K with a 25-K step. As depicted in Figure 4b, after 100 K, the SS for the GaN on Si HEMT increased by 1.32 times, whereas that of the GaN on the QST device increased by 1.14 times. Between 300 and 400 K, the leakage current of the QST device ranged from 2.9 × 10^−5^ to 6.1 × 10^−5^ mA/mm and that of the Si device ranged from 1.4 × 10^−6^ to 7 × 10^−6^ mA/mm. The leakage current variation of the QST device was lower than that of the Si device and nearly twice at *V*_GS_ = −6 V because the QST substrate has a high thermal conductivity, which enables it to disperse heat effectively. As presented in Figure 4c, the maximum saturation currents of the Si device at 300 and 400 K were 538 and 396 mA/mm, respectively, at a *V*_DS_ of 10 V. The external temperature affected the device current, which decreased by nearly 26.5% as the temperature increased from 300 to 400 K. At 300 and 400 K, the currents of the QST device were 547 and 451 mA/mm, respectively, at a *V*_DS_ of 10 V; thus, the current decreased by almost 17.5% with an increase in the temperature from 300 to 400 K. The *I*_DS_ degradation of the GaN on QST (17.5%) was 9% smaller than that of the GaN on Si (26.5%) when the devices were operated in a high-temperature environment (400 K), as shown in Figure 4d. The normalized *R*_ON_ ratio of the GaN on QST and Si increased by 1.32 and 1.57 times, respectively, as the temperature increased from 300 K to 400 K. The results depicted in Figure 4 indicate that the QST substrate can be used at relatively high temperatures because of its high thermal conductivity.

Temperature is a crucial consideration for device reliability. We measured the surface temperature distribution in both devices under operation at a high current and observed the self-heating effect of the different substrates. The surface temperature maps presented in Figure 5a,b were obtained according to the infrared radiation intensity measured by an IRM P384G detector. They determined the emissivity calibration of the QST and Si substrate devices within 50 s of operation at a current of 300 mA/mm. As illustrated in Figure 5a,b, the surface peak temperatures of the Si device ranged from 36.9 °C to 46.1 °C and those of the QST device ranged from 36.6 °C to 41.1 °C. Because the GaN on QST dissipated heat quickly, the temperature outside the operating device did not increase considerably. The heat of the GaN on Si did not dissipate; however, after 22 s of device operation, the temperature detected using the thermal imager for the GaN on Si was higher than that of the GaN on QST. Because of the slow heat dissipation of the Si substrate, heat accumulated in the buffer layer and substrate and diffused to the area outside the element. This phenomenon caused the temperature of the area outside the operating element to approach that of the operating element itself when the thermal imager was measuring the temperature. As illustrated in Figure 5c, at the end of the operation, after approximately 60 s, the device naturally cooled down. Compared with the GaN on Si, the GaN on QST exhibited a higher temperature decrease. The results revealed that self-heating affected the QST substrate less than the Si device. Because the thermal conductivity of the GaN on the QST device was higher than that of the GaN on the Si device, the heat dissipation performance of the GaN on the QST device was superior, with the device operating at an *I*_DS_ of 300 mA/mm and a *V*_DS_ of 10 V.

## 4. Conclusions

This study explored the characteristics of AlGaN/GaN HEMTs grown on QST and Si substrates. Because of the material characteristics of the QST substrate, the GaN on this substrate had more heat dissipation than did the GaN on a Si substrate. Measurement and analysis results indicated that compared with the GaN on the Si substrate, the GaN on the QST substrate had a higher heat dissipation rate. Moreover, the effect of device operation at high temperatures was weaker for the GaN on the QST substrate than for the GaN on the Si substrate; thus, GaN on QST substrates is more suitable for high-temperature operations than is GaN on Si substrates. The high-thermal-conductivity QST substrate not only enabled the device to operate stably in a high-temperature environment but also exhibited strong performance in terms of the self-heating effect. The effective heat dissipation characteristic of this substrate indicates the potential of engineered substrates as effective RF platforms for 5G microcell or macrocell base stations.

## Figures and Tables

**Figure 1 membranes-11-00848-f001:**
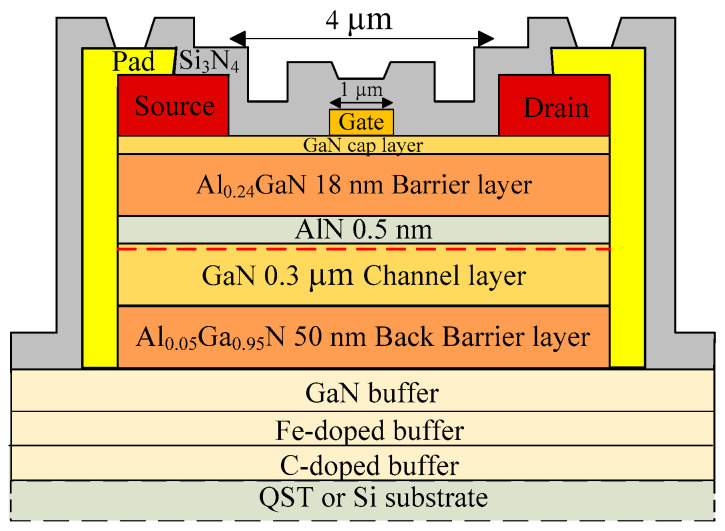
Cross-sections of the GaN on QST, with *L_G_*, *L_DS_*, and *W_G_* being 1, 4, and 50 µm, respectively.

**Figure 2 membranes-11-00848-f002:**
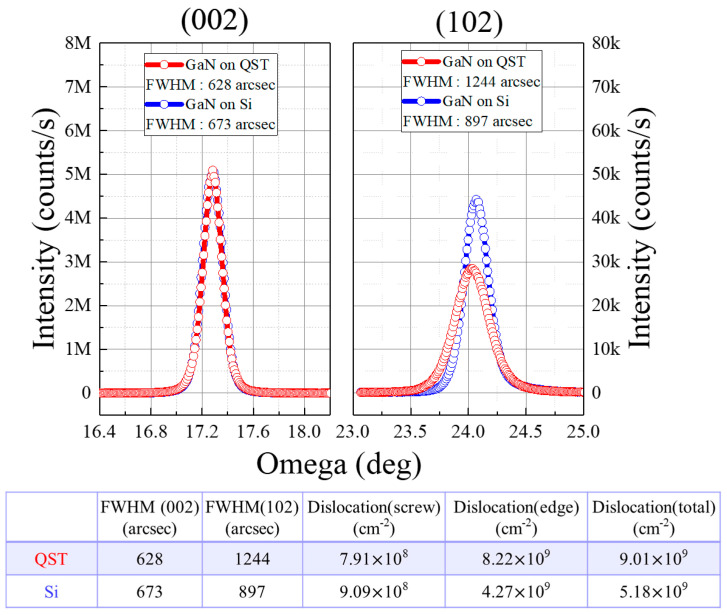
XRD results and total dislocation of the GaN on QST and Si.

**Figure 3 membranes-11-00848-f003:**
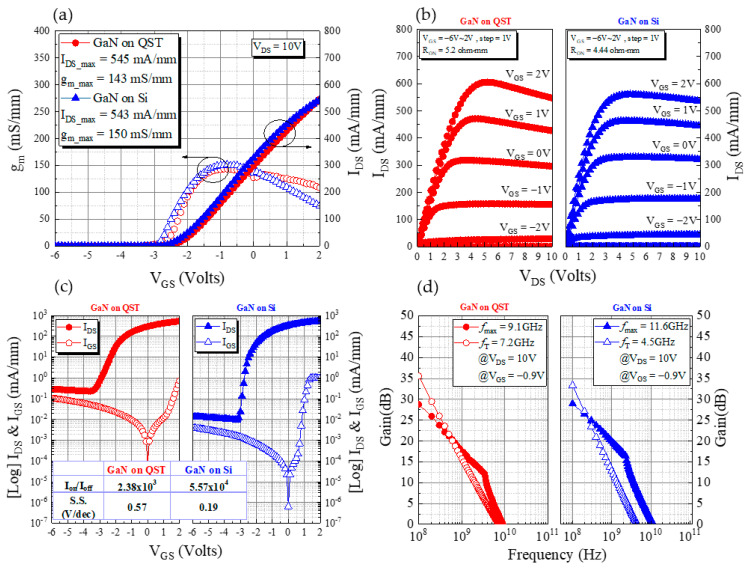
(**a**) Transfer characteristics (*I*_DS_–*V*_GS_) at *V*_DS_ = 10 V with a *V*_GS_ sweep of −6 to 2 V, (**b**) *I*_DS_–*V*_DS_ output current at a *V*_GS_ sweep of −6 to 2 V with a step of 1 V, (**c**) off-state leakage current curves of the drain (log-scale *I*_DS_–*V*_GS_) and gate (*I*_GS_–*V*_GS_), and (**d**) small-signal characteristics of the QST and Si devices.

**Figure 4 membranes-11-00848-f004:**
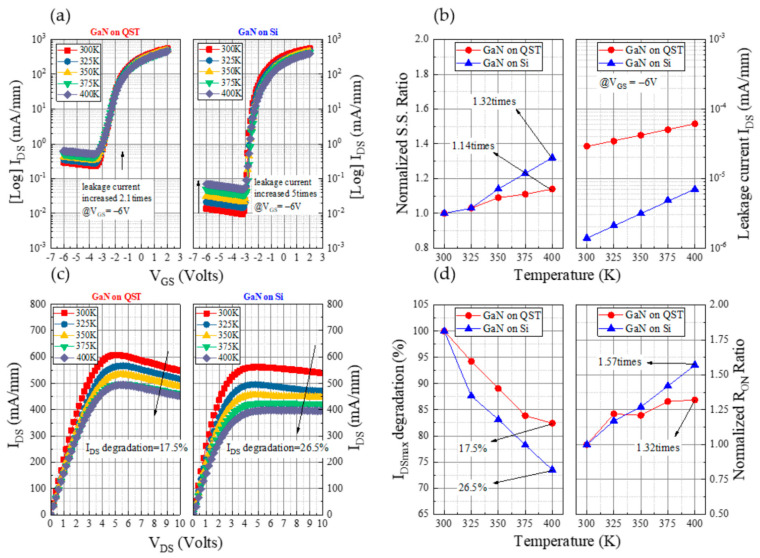
Analysis under various temperatures: (**a**) Transfer characteristics (*I*_DS_–*V*_GS_) for GaN on Si and GaN on QST, (**b**) SS ratio and leakage current at *V*_GS_ = −6 V, (**c**) *I*_DS_–*V*_DS_ output current, and (**d**) *I*_DSmax_ degradation and *R*_ON_ ratio.

**Figure 5 membranes-11-00848-f005:**
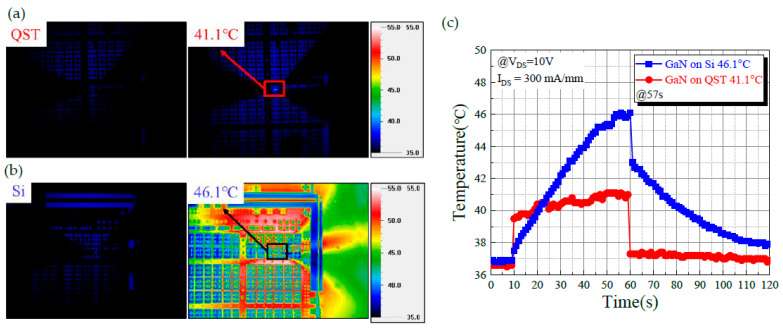
Analysis of the thermal image measurements for (**a**) GaN on QST and (**b**) GaN on Si, and (**c**) The analysis of the thermal behavior at an *I*_DS_ of 300 mA/mm.

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
