# Peer review of "Characteristic Analysis of AlGaN/GaN HEMT with Composited Buffer Layer on High-Heat Dissipation Poly-AlN Substrates"

_membranes, 2021, doi:10.3390/membranes11110848_

Round 1
Reviewer 1 Report
The authors have compared the DC and RF characteristics of AlGaN/GaN HEMTs on poly-AlN substrates with those on Silicon substrates, with particular emphasis on the devices heat dissipation performance. The presented work could contribute to GaN high power and/or high frequency research field. However, several issues outlined below should be addressed before publication:
1) The authors concluded that:
“The high-thermal-conductivity QST substrate not only enabled the device to operate stably in a high-temperature environment but also exhibited a good performance in the self-heating effect.”
However, from the comparison of drain curves shown in Figure 3 (b), the device on QST showed higher negative slope of the ID-VDS curve at VGS=2 V in saturation regime. The higher this negative slope, which is also known as “negative differential resistance” (J. Kuzmik et al., IEEE TED 49 (2002) 1496), the higher the degree of self-heating in the device (J. T. Asubar et al., APL 105 (2014) 053510). Why is this so?
2) The acronyms QST and BB should be spelled out at first mention.
3) Where is the equality sign of equation (1) given on page 2? Could the authors provide a reference for this equation?
4) The authors should discuss the reason for the much worse subthreshold swing and drain current on-to-off ratio values (Figure 3 (c)) of the device on QST compared with those of the device on Si substrate. Is it only due to much higher number of dislocations of the device on QST?
5) What is the mechanism behind apparent increase in ft (Figure 3(d)) of the device on QST over the device on Si?
6) I recommend that the authors should have their paper checked by a native English speaker.
Author Response
Reviewer 1
1. “The high-thermal-conductivity QST substrate not only enabled the device to operate stably in a high-temperature environment but also exhibited a good performance in the self-heating effect.”
However, from the comparison of drain curves shown in Figure 3 (b), the device on QST showed higher negative slope of the ID-VDS curve at VGS=2 V in saturation regime. The higher this negative slope, which is also known as “negative differential resistance” (J. Kuzmik et al., IEEE TED 49 (2002) 1496), the higher the degree of self-heating in the device (J. T. Asubar et al., APL 105 (2014) 053510). Why is this so?
Ans: Thanks for your suggestion. In this work, because of the higher dislocation density of QST device, it affects the self-heating effect of the device at higher drain bias. This phenomenon can also be seen in Figure 5(c), when the device was operated between 10 to 20 seconds. However, due to the excellent thermal conductivity of QST, it quickly disperses and reduce the temperature of the device self-heating effect in 20 to 60 seconds. From this result, if we can improve the defects caused by epitaxy, we will be able to have a better performance in the self-heating effect.
The acronyms QST and BB should be spelled out at first mention.
Ans: Thanks for your suggestion and we have corrected it in the revised version.
line 14: Qromis' substrate technology (QST)
line 42: back barrier layer (B.B.)
Where is the equality sign of equation (1) given on page 2? Could the authors provide a reference for this equation?
Ans: Thanks for your suggestion and we have corrected the equation [16,17] in the revised version.
[16] N.V. Safriuk, G.V. Stanchu, A.V. Kuchuk, V.P. Kladko, A.E. Belyaev, V.F. Machulin, “X-ray diffraction investigation of GaN layers on Si (111) and Al2O3 (0001) substrates”, Semicond. Phys. Quantum Electron. Optoelectron. doi:10.15407/spqeo16.03.265.
[17] I. Booker, L. R. Khoshroo, J. F. Woitok, V. Kaganer, C. Mauder, H. Behmenburg, J. Gruis, M. Heuken, H. Kalisch, and R. H. Jansen, “Dislocation density assessment via X-ray GaN rocking curve scans”, Physica Status Solidi (C), 7, No. 7–8, 1787–1789, 2010, doi:10.1002/pssc.200983615.
The authors should discuss the reason for the much worse subthreshold swing and drain current on-to-off ratio values (Figure 3 (c)) of the device on QST compared with those of the device on Si substrate. Is it only due to much higher number of dislocations of the device on QST?
Ans: Thanks for your careful reading. Yes, the higher dislocation affects the subthreshold swing of the device and this result can also be observed from the increased Ron characteristic in the figure 3 (b).
What is the mechanism behind apparent increase in ft (Figure 3(d)) of the device on QST over the device on Si?
Ans: Thanks for your careful reading. Because of the higher leakage current of QST device in Figure 3 (c), the cut-off frequency (fT) may extra gain from leakage current. With proper drain bias point, the QST device performed a better current density with the same gate swing voltage, thus the gm and related ft can be improved.
I recommend that the authors should have their paper checked by a native English speaker.
Ans: Thanks for your suggestion and the English editing has been submitted and polished by a native English speaker.

Reviewer 2 Report
Dear authors, please find my comments in the following.
Best regards.
In order to improve the readability, consider to insert in Figure 1 also the structure of the GaN HEMT on SI.
Please give a reference for equation 1.
Line 115-131. Please correct the reference to Fig5 with Fig4.
Line 127, In order to improve the readability, please insert a reference to Figure 4d.
Regarding Figure 5:
The fact that the device is in the entire wafer can falsify the obtained thermal results.
Consider to comment the temperature results since the device is on an entire wafer and the temperature results are not good representative values for the device when it is inserted in the final package.
Author Response
- In order to improve the readability, consider to insert in Figure 1 also the structure of the GaN HEMT on SI.
Ans: Thanks for your suggestion and we have corrected the structure the of Figure 1.
Please give a reference for equation 1.
Ans: Thanks for your suggestion and we have added the reference [16] and [17].
[16] N.V. Safriuk, G.V. Stanchu, A.V. Kuchuk, V.P. Kladko, A.E. Belyaev, V.F. Machulin, “X-ray diffraction investigation of GaN layers on Si (111) and Al2O3 (0001) substrates”, Semicond. Phys. Quantum Electron. Optoelectron. doi:10.15407/spqeo16.03.265.
[17] I. Booker, L. R. Khoshroo, J. F. Woitok, V. Kaganer, C. Mauder, H. Behmenburg, J. Gruis, M. Heuken, H. Kalisch, and R. H. Jansen, “Dislocation density assessment via X-ray GaN rocking curve scans”, Physica Status Solidi (C), 7, No. 7–8, 1787–1789, 2010, doi:10.1002/pssc.200983615.
Line 115-131. Please correct the reference to Fig5 with Fig4
Ans: Thanks for your suggestion and we have corrected the reference to Fig 5 with Fig 4.
Line 127, In order to improve the readability, please insert a reference to Figure 4d.
Ans: Thanks for your suggestion and we have inserted a reference to Figure 4d.
“The IDS degradation of GaN on QST (17.5%) was smaller than that of GaN on Si (26.5%) at 9% when the device was operated in a high-temperature environment (400 K), shows in Fig. 4 (d).”
Regarding Figure 5:
The fact that the device is in the entire wafer can falsify the obtained thermal results.
Consider to comment the temperature results since the device is on an entire wafer and the temperature results are not good representative values for the device when it is inserted in the final package.
Ans: Thanks for your suggestion. Although the heat dissipation characteristics of the device will be greatly improved after packaging, the heat dissipation of the substrate will also have an impact, so we use thermal image analysis to initially verify the heat dissipation characteristics of the device.
